# A Recyclable Polypropylene Multilayer Film Maintaining the Quality and the Aroma of Coffee Pods during Their Shelf Life

**DOI:** 10.3390/molecules29133006

**Published:** 2024-06-25

**Authors:** Martina Calabrese, Lucia De Luca, Giulia Basile, Gianfranco Lambiase, Raffaele Romano, Fabiana Pizzolongo

**Affiliations:** 1Department of Agricultural Sciences, University of Naples Federico II, Via Università 100, Portici, 80055 Napoli, Italy; martina.calabrese@unina.it (M.C.); lucia.deluca@unina.it (L.D.L.); giulia.basile@unina.it (G.B.); raffaele.romano@unina.it (R.R.); 2Flessofab s.r.l., Montemiletto, 83038 Avellino, Italy; lambiase@flessofab.it

**Keywords:** packaging, sustainability, oxidation, peroxides, volatile organic compounds, oriented polypropylene, oxygen in headspace

## Abstract

Films for coffee-pod packaging usually contain aluminium as an impermeable foil that is not recyclable and has to be discharged as waste. In this study, a recyclable polypropylene multilayer film is proposed as an alternative. The performance on the chemical composition of coffee was evaluated and compared to that of film containing aluminium (standard). The oxygen in the headspace, moisture, lipidic oxidation, and volatile organic compounds were studied in coffee pods during storage for 12 months at 25 and 40 °C. In addition, the acidity and acceptability of extracted coffee were evaluated. In the polypropylene-packaged pods, the percentage of oxygen during storage at 25 °C was lower than that in the standard. Moisture was not affected by the type of packaging materials. No differences were found between the peroxide values, except in pods stored for 3, 10, and 11 months at 25 °C, where they were even lower than the standard. Furans and pyrazines were the main volatile organic compounds detected. No differences were found in the pH and titratable acidity of the coffee brew either. All samples were well accepted by consumers without any perceived difference related to the packaging film. The polypropylene multilayer film is a sustainable recyclable material with high performance, in particular, against oxygen permeation.

## 1. Introduction

Coffee is among the most commonly consumed beverages in the world. The *Coffea arabica* variety represents 70% of the world’s coffee production, is generally considered of higher quality, and is more expansive than *Coffea canephora* var. *robusta* [1,2].

Consumer preference towards innovation, convenience, and lifestyle changes led to a rapid growth in the coffee-pod market. Coffee pods are a single dose of ground roasted coffee pressed in round or flat paper packages and packed under a protected atmosphere in impermeable foil. They allow for the preparation of a relatively good coffee very quickly and are in great demand by consumers. Their disadvantage is that they are usually packaged in impermeable foils containing aluminium foil that is not sustainable for the environment and has to be discharged as waste. Nowadays, the coffee-pod industry is trying to use alternative and more sustainable packaging to avoid this waste that represents a critical issue [3]. From a sustainable point of view, aluminium should be replaced with compostable or at least recyclable materials.

As reported by several studies in the literature [4,5,6], the materials most commonly used to package coffee pods are a polyethylene terephthalate (PET) metalized multilayer, polystyrene (PS), or oriented polypropylene (PP) and an aluminium multilayer [7]. Among recyclable materials, PP is the most used for its cheapness and mechanical properties. It is suitable for foodstuff packaging, with a thickness of <100 mm, and has good degradability among polyoleofins [8]. PP is a linear polymer of hydrocarbons that contains little or no unsaturation. Currently, it is one of the most widely used thermoplastic materials. Its great popularity is due to the advantageous combination of a low cost and a wide range of properties [9]. PP films are produced by extrusion and are divided into two broad categories: cast films and oriented films. Cast films are manufactured by depositing a layer of liquid plastic onto a surface and solidifying it by cooling the melt or by the evaporation of a solvent. Its main feature is softness [10]. In order to improve the mechanical properties, PP, as any polymer, both amorphous and crystalline, can be subjected to orientation, increasing its mechanical strength and becoming less permeable [11]. A biaxially oriented polypropylene film is a type of film produced by extruding the plastic through a circular mold, followed by a cooling and expansion process. This type of film offers greater water vapor resistance, keeping packaged products stable for longer [10]. Moreover, highly oriented polypropylene films were consolidated to create a fully recyclable, high performance all-polypropylene composite [12]. The main issue related to PP is its poor oxygen barrier properties. To overcome this drawback, some companies use a layer of ethylene vinyl alcohol sandwiched between two PP layers or other polylaminate films [13]. Although PP films are recyclable, many of these are not recycled, because they consist of multiple films that are difficult to separate [7].

Coffee aroma is influenced by the packaging material during the shelf life, because highly volatile compounds can be lost. In addition, oxygen can permeate inside the coffee pods and oxidate the molecules, resulting in a loss of freshness [14]. The volatilization of aroma molecules can be limited by protecting the coffee with impermeable packaging. The oxidation of aroma compounds can be prevented by packaging the coffee in a protective atmosphere inside the pack. The main physical and chemical events involved in the formation of the negative stale taste of roasted coffee during storage are the release of volatile compounds and carbon dioxide, the migration of oils to the surface, and oxidation reactions. Volatile organic compounds (VOCs) are responsible for the aroma of coffee. Green coffee (unroasted coffee) has more than 200 VOCs that are of limited importance to the aroma [15]. During roasting, chemical reactions, such as the Maillard reaction, Strecker degradation of amino acids, pyrolysis, and others, produce a large number of different VOCs. So far, more than 800 different compounds have been identified in roasted coffee. They are from a wide range of chemical classes, and most of them can be grouped in aldehydes, which are responsible for fruit aroma, furans (caramel aroma), pyrazines (musty), and phenols (spices, smoked) [16].

The VOC barrier property of packaging material is an important parameter. VOC penetration through a polymeric film is affected by several factors, such as the chemical composition of the polymer, the type and concentration of the VOC, the temperature, the relative humidity, storage conditions, etc. Previous studies [17] showed that the ester VOCs of fruits (orange and apple juices) are absorbed by low-density polyethylene and PP more than PET films. Studies in model solutions were also carried out, showing that hydrophobic aroma compounds have an affinity for and interact with non-polar polymers, such as low-density polyethylene.

Various studies in the literature describe how coffee’s aroma profile changes depending on the material used for packaging, emphasizing the importance of the barrier properties of materials on aroma retention. For example, Trenzová et al. [18] examined the trend of volatile organic compounds during the secondary shelf-life of different coffee packages subjected to repeated opening and closing, simulating gradual consumption by the consumer. Additionally, Gloess et al. [19] evaluated some indices of the freshness of coffee stored in single-serving capsules. The evolution of VOCs is closely related to the barrier properties of packaging, both directly and indirectly. There may be a direct exchange of molecules between the headspace and the external environment due to the permeation of the packaging, leading to a loss of certain molecules in the coffee volatilome and, consequently, a loss of aroma. Alternatively, there may be an indirect evolution of the volatilome due to oxygen permeation, primarily, and subsequent chemical reactions, such as lipid oxidation, which result in the formation of secondary compounds that also contribute to coffee flavour.

Also, lipids may influence the sensory quality, especially the body of a coffee beverage. Their content in roasted *Coffea arabica* is approximately 11–20% [20]. Triacylglycerols, the main lipid class (75% of total coffee lipids), can be hydrolysed at different rates in the function of moisture and temperature during storage. The free fatty acids released during the hydrolysis are oxidized to produce off-flavours in coffee.

Despite the worldwide importance of coffee pods, studies on their chemical composition during their shelf life in the function of the packaging used are scarce. According to Illy and Viani, [21] ground coffee contained in cans containing inert gas and with a residual oxygen percentage of 1–2% has a shelf life of 6–8 months before an alteration.

With the goal to promote sustainable packages, in the present work, we used a film made exclusively of PP to package coffee pods and studied the evolution of the aroma and chemical composition of coffee during storage. The recyclable multilayer PP film (REC) was composed of 15 microns of oriented polypropylene high performance, 16 microns of oriented polypropylene high barrier, and 50 microns of a cast polypropylene. The performance on a 100% *Coffea arabica* chemical composition was evaluated and compared to that of a standard, non-recyclable aluminium-containing film (STD). In particular, the oxygen in the headspace, lipidic oxidation, and volatile organic compounds were studied in the coffee pods during storage for 12 months at 25 and 40 °C. In addition, the acidity (pH and titratable acidity) and acceptability of the extracted coffee were also evaluated.

## 2. Results and Discussion

### 2.1. Oxygen in Coffee Pod Headspace

The overall trend in the oxygen permeation across STD and REC films is displayed in Figure 1. During the first month of storage at 25 °C, the percentage in the headspace of the coffee pods decreased from 0.5 to 0.3%, both in STD- and REC-packaged samples, without any statistically significant difference between the two packaging. The decrease can be justified, because just after roasting, coffee has a remarkably negative redox potential value, indicating its strong reducing properties [22]. For this reason, oxygen is consumed in oxidation reactions.

Oxygen slowly entered the headspace, achieving a maximum of 0.93% after 12 months of storage at 25 °C both in REC and STD films. According to Nicoli et al. [22], a final percentage of oxygen of 1–2% allows for a shelf life of up to 6–8 months. From the second month onwards, the percentage of oxygen is significantly lower in the headspace of pods packaged in a REC film than in that of those packaged in STD, indicating a better performance than the first film. A similar trend was observed in accelerated storage at 40 °C, with the exception of data measured after 12 months, which were significantly lower in the REC samples.

### 2.2. Moisture in Coffee Pods

Moisture gain is a critical parameter that affects the quality and shelf life of foods with low moisture. Because of the high temperature during the roasting process, coffee is characterized by a very low water content. Moisture after roasting controls aroma retention and stability during the storage of coffee [23]. The moisture content of coffee pods significantly increased during storage from an initial of 1.15–1.21% to 2.11–2.51% (Table 1). This result is in good agreement with the work of Agustini and Yusya [24] on ground roasted coffee. Increased moisture could lead to favourable conditions for microbial growth. The type of packaging materials had no significant effect on the moisture content, demonstrating that the REC film exhibited protective barriers as high as those of aluminium foil. The same trend was observed at 25 and 40 °C. The moisture content of all pods was acceptable, since it was less than 5%, the maximum value at the time of packaging, according to international standards for the quality of roasted and ground coffee [25].

### 2.3. Fat and Peroxide Values in Coffee Pods

The average fat content in coffee pods was 10.85%, without statistically significant differences among the samples. These values are similar to those reported by Rubayiza and Meurens [26], who found 16.8% fat on a dry-matter basis in 100% arabica coffee. Fat can undergo an oxidative process in the presence of oxygen. The rancidity of coffee pods was controlled by an analysis of the peroxide value (PV). Peroxides are primary products of the oxidative process and have been used as an oxidative index in roasted coffee by other authors [27]. In Figure 2, the PV of coffee pods stored at 25 °C (A) and 40 °C (B) in the two types of packaging are reported. During the first three months of storage, all samples showed a PV of below 2 meq O_2_/kg of oil at both temperatures. Similar results were found in our previous work [28], where PVs were determined in pods packaged in a PET metalized film. From the third month onwards, the PV significantly increased up to 10 meq O_2_/kg of oil. This is correlated with the progressive ingress of oxygen into the headspace package detected previously. Coffee pods in the REC film showed a PV that was significantly lower than that in STD at 3, 10, and 11 months of storage (a reduction of 39.5%, 10.8%, and 13.5%, respectively), while at the other storage times, no significant differences were found. The PV in samples at accelerated storage (40 °C) were higher than those in samples at 25 °C, but no significant differences between the two types of film were found. We can conclude that in a REC film, oxidation processes occur to a lesser extent due to its better barrier properties towards oxygen.

### 2.4. Volatile Organic Compounds in Coffee Pods

In order to study the loss of the aroma of coffee in the two types of packaging during a shelf life, volatile organic compounds (VOCs) were analysed. Coffee volatilome is very complex, and, generally, volatile organic compounds are mostly generated by thermal reactions during roasting, such as the Maillard reaction, Strecker degradation, and pyrolysis [16]. In Table 2 and Table 3, the relative percentage of VOCs in coffee pods during storage in STD and REC films at 25 °C and 40 °C, respectively, is reported.

Furans are the main components, with a value of 43% at t0. They are formed during the roasting of coffee by the thermal degradation of carbohydrates alone or in the presence of amino acids, by the thermal degradation of some amino acids, by the oxidation of ascorbic acid at elevated temperatures, and by the oxidation of polyunsaturated fatty acids and carotenoids [29]. Due to the high volatility of furan, any subsequent processing step or consumer handling has an impact on the level of furan. It is estimated that only approximately 10% of the initially generated furan during roasting gets into the cup of coffee for consumption [30]. The furan content is significantly lower in pods inside REC than in that of those in STD after 4, 8, and 12 months of storage at 25 °C. However, the level is in the range of 32–37%, which is not far from the range of pods in STD (35–41%). The furan content significantly decreased during storage in the REC film at both temperatures, reaching a value of 25 and 33% after 12 months at 25 and 40 °C, respectively. The main furan derivate detected was 2-furanmethanol at 28%.

Total pyrazines were present at the 30% level at t0, and 2,5 dimethylpyrazine was the most abundantly detected. Pyrazines and thiols were found, in most cases, as the most potential odour compounds formed in the Maillard reaction [31]. They are present in coffee at levels exceeding their threshold value and, therefore, have a high odour value. 2,5 dimethylpyrazine has a nutty, grassy odour descriptor and a 18–35 ppm odour threshold in water [32]. No significant differences between pods stored in STD and REC films were found.

The total phenols were 7.9% of the total VOCs, and 2-methoxyphenol was the most abundant. This compound is negatively associated with liking, because it is perceived as bacon, medicine, and the like [33]. Phenols are generated by the degradation of chlorogenic acid that occurs during roasting [34]. No significant differences between pods stored in STD and REC films were found.

Pyridines were detected at 4.8%. These compounds are produced through the degradation of trigonelline and are characterized by an unpleasant, plant-like, and bitter odour that signals their presence in products. In particular, 2-methylpyridine has a hazelnut odour and 3-ethylpyridine a buttery, plant-like, and caramel aroma [35]. No significant differences between pods stored in STD and REC films were found, except at a t of 4 months, when the content in the pods in STD was higher than those in REC.

Ketones were found at 6%, with 1-(acetyloxy)-2-propanone being the most abundant. They are associated with positive notes [36]. No significant differences between pods stored in STD and REC films were found.

Organic acids represented 5% of VOCs, and acetic acid was 4.5%. The samples stored in the REC film showed a percentage of acetic acid that was significantly higher than those stored in STD. In both types of samples, the percentage increased during the storage by up to 15% in the sample stored for 12 months at 40 °C. Generally, the consumers’ preference is strongly correlated with acidity. Acetic acid is associated with sourness, as well as rancidity, astringency, and bitterness [37]. Organic acids are not crucial to coffee aroma. Their contribution could be either positive with cheese, cream, and chocolate notes, or negative with sweat-like notes [38].

### 2.5. pH and Titratable Acidity of Extracted Coffee

In order to verify if a pod’s packaging can affect the chemical composition of a coffee brew, the pH and titratable acidity were measured in the extracted coffee. Measurements of pH quantify the concentration of deprotonated acid molecules, measuring the concentration of hydrogen ions in an aqueous solution. Total titratable acidity measures all acidic protons, including non-dissociated protons, that are neutralized through the addition of a strong base during a titration analysis. The main acids in a coffee brew are citric, malic, pyruvic, succinic, quinic, acetic, and formic [39,40], but their composition depends on the variety of Coffea, roasting conditions, and the method of extraction of the coffee brew.

The pH values of a coffee brew obtained from pods stored in STD and REC films are reported in Table 4. The values are in the range of 5.66–5.28 and are in agreement with those found by Pérez-Martínez et al. [41]. The pH decreased during the storage from 5.66 to 5.32 and 5.28 at 25 and 40 °C, respectively, indicating an increase in the acid content in pods during storage. No significant differences between samples from pods in STD and REC films were recorded.

As the higher value of pH indicates the lower acidity of a solution, the titratable acidity followed an opposite trend with respect to pH during storage (Figure 3). The acidity was in the of range 2–5 mg NaOH/mL. No significant differences between samples from pods in STD and REC films were recorded at 25 °C. The acidity was lower in a coffee brew from pods stored in REC than those in STD after 1 and 10 months of storage at 40 °C.

### 2.6. Acceptability of Extracted Coffee

An acceptability test was performed on the extracted coffee to detect if packaging influences the sensory characteristic of a coffee brew. In Table 5, the smell and taste acceptability of the coffee brew is reported. In general, samples were well accepted by the consumers until 12 months of storage at 25 °C. The scores were of approximately 8 corresponding to “Like very much” and 7 corresponding to “Like very much—Like moderately” in samples from pods after 12 months of storage. The increase in peroxide and acetic acid values during pod storage did not affect the overall acceptability of the coffee brew. No differences regarding the film used for packaging were perceived.

## 3. Materials and Methods

### 3.1. Chemicals

All solvents and reagents used for the experiments were purchased from Sigma-Aldrich Co. (Milano, Italy).

### 3.2. Samples

Coffee pods were provided by Kimbo Spa (Naples, Italy). Each pod was made of paper and contained 8.5 g of 100% *arabica* coffee, medium roast (230 °C for 12 min). The pods were individually sealed within a protective atmosphere consisting of 100% nitrogen (N_2_) and were packaged in two different types of films: standard (STD) non-recyclable aluminium-containing and recyclable (REC) films. Both films were purchased from Flessofab srl (Montelimetto, Avellino, Italy).

The STD film consisted of 12 microns of polyethylene terephthalate (PET), 8 microns of aluminium, and 60 microns of a polyethylene (PE) layer, with an oxygen transmission rate (OTR) of <1.0 and a grammage of 99 g/m^2^.

The REC film, without aluminium, consisted of a high-barrier polypropylene film composed of 15 microns of oriented polypropylene high performance (OPPHP), 16 microns of oriented polypropylene high barrier (OPPHB), and 50 microns of a cast polypropylene (CPP/B) layer, with an OTR of <0.5 and a grammage of 81 g/m^2^.

Considering the grammage, the REC film resulted in an 18% reduction in packaging weight.

Pod samples were stored for 12 months at 25 °C and at 40 °C to simulate accelerated storage conditions. Monthly measures of the percentage of oxygen and peroxide values were performed, while the moisture content and volatile organic compounds (VOCs) were determined at 0, 4, 8, and 12 months.

### 3.3. Percentage of Oxygen during Shelf Life

During the shelf life tests, the oxygen percentage (%) in the headspace of the package was analysed with a Witt Oxybaby 4.0 analyser before opening.

### 3.4. Moisture Determination

For the determination of the moisture content, approximately 3 g of coffee powder was dried in an oven at 105 °C for 24 h. The results are expressed as a percentage weight/weight (% *w*/*w*).

### 3.5. Fat Extraction

Fat extraction from the coffee was carried out following the method described by Cong et al. [27], with some modifications. In brief, 20 g of coffee powder was mixed with 100 mL of n-hexane in a 50 mL flask at room temperature (25 °C) in an ultrasonic bath for 50 min. The sample was then centrifuged at 6500 rpm for 10 min. The organic phase was collected, and the solvent was evaporated using a Rotavapor Laborota 4000 Efficient instrument (Heidolph Instrument, Schwabach, Germany). The extraction oil was stored at −18 °C until further analysis. The fat extraction yield is expressed as g of oil extracted from 100 g^−1^ coffee.

### 3.6. Determination of Peroxide Value (PV)

The PV was determined using iodometric titration according to the official method from AOAC [42]. Approximately 1 g of coffee oil was mixed with 10 mL of a solution containing acetic acid and chloroform in a 3:2 (*v*/*v*) ratio. Then, 0.1 mL of a saturated KI solution was added, and the mixture was allowed to react in darkness for about 5 min. After this time, 15 mL of deionized water was added to halt the reaction, along with starch salt as an indicator. The solution was titrated with 0.001N of sodium thiosulfate (Na_2_S_2_SO_3_) until the indicator-induced coloration completely disappeared. Results are expressed as meqO_2_ per kg^−1^ of coffee oil.

### 3.7. Volatile Organic Compounds (VOCs)

The analysis of VOCs was performed using the solid phase microextraction (SPME) technique coupled with gas chromatography (GC/MS), according to Bertrand et al. [43]. A quantity of 2 g of coffee powder was taken from each pod. The SPME and GC/MS conditions were the same as those used by Basile et al. [21]. The NIST (National Institute of Standards and Technology, Gaithersburg, MD, USA) Atomic Spectra Database version 2.0 was used to identify the analytes. The relative coffee VOC content was calculated from peak area ratios and is expressed as a percentage (%).

### 3.8. pH and Titratable Acidity in the Extracted Coffee

Each sample of coffee pods was introduced in a Mokona machine (Bialetti Industri, S.p.a, Coccaglio, BS, Italy) to obtain approximately 30 mL of the aqueous extract. Coffee brew samples were rapidly cooled at 20 °C, and the pH and the titratable acidity were measured. The pH value, which represents the hydrogen ion concentration, was determined by a pH meter (Medidor pH BASIC 20 Crison Instruments, Barcelona, Spain). The titratable acidity was determined following the method of Gloess et al. [44], while making necessary modifications. After coffee extraction, the sample was cooled, and then 2 mL of the extract was mixed with 100 mL of deionized water. This mixture was titrated with 0.1 N of NaOH in the presence of phenolphthalein as an indicator until a persistent pink colour was obtained for 30 s. The titratable acidity is expressed as mg of NaOH per mL of coffee extract (mg of NaOH mL^−1^ of coffee extract).

### 3.9. Sensory Evaluation of Extracted Coffee

The overall acceptability of the extracted coffee brew, as previously described, was evaluated. A semi-structured hedonic scale was used [45]. The acceptability test was performed as reported by Basile et al. [28], with some modifications. The overall acceptability of the smell and taste of the brewed coffee were evaluated. A nine-point hedonic scale was used by panellists. “Like Extremely” and “Dislike Extremely” were at either end of the scale, and “Neither Like nor Dislike” was in the middle as a neutral point.

### 3.10. Statistical Analysis

All analyses were repeated three times, and the results are reported as an average ± standard deviation. The data were statistically evaluated by means of a one-way analysis of variance (ANOVA) and Tukey’s test (*p* ≤ 0.05) using the XLSTAT software version 2023 (Addinsoft, New York, NY, USA).

## 4. Conclusions

The recyclable polypropylene multilayer film showed a better performance than the standard film in the packaging and storage of coffee pods. It exhibited a good barrier against oxygen, moisture, and oxidation during storage for 12 months. The volatile organic compounds of coffee were not affected by the type of packaging materials, and furans and pyrazines were mainly detected. No differences were found in the pH and titratable acidity of the coffee brew either. All samples were well accepted by consumers without any perceived difference related to the packaging film. Therefore, the polypropylene multilayer film proposed to package coffee pods is a valid and sustainable alternative to the standard aluminium-based film. Furthermore, considering its lower grammage (81 vs. 99 g/m^2^), it allows for a reduction in the volume of packaging to waste.

## Figures and Tables

**Figure 1 molecules-29-03006-f001:**
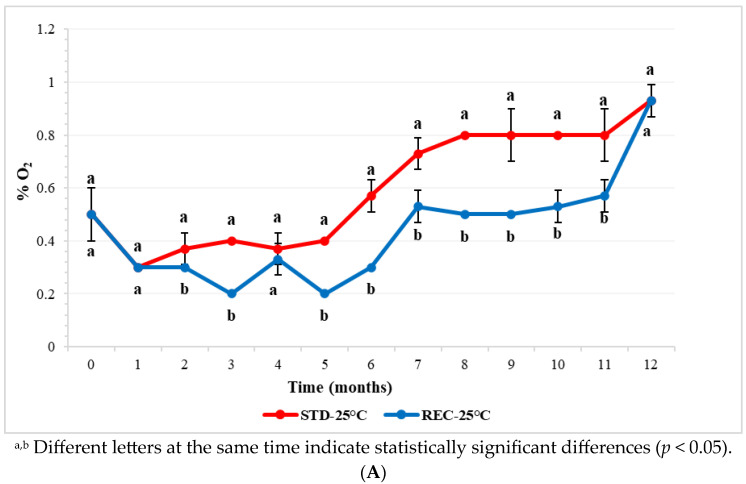
Percentage of oxygen in the headspace of coffee pods packaged with standard (STD) and recyclable (REC) multilayer films and stored at 25 °C (**A**) and 40 °C (**B**) for 12 months.

**Figure 2 molecules-29-03006-f002:**
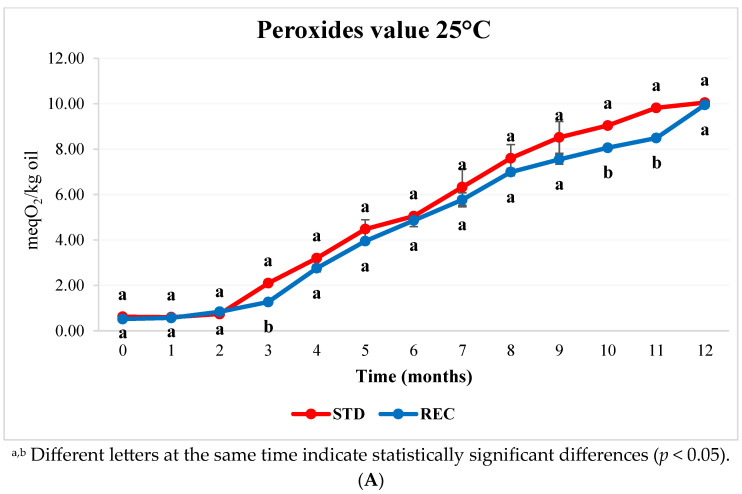
Development of peroxide values (meq O_2_/kg of oil) in coffee pods packaged with standard multilayer film (STD) and recyclable film (REC) and stored at 25 °C (**A**) and 40 °C (**B**) for 12 months.

**Figure 3 molecules-29-03006-f003:**
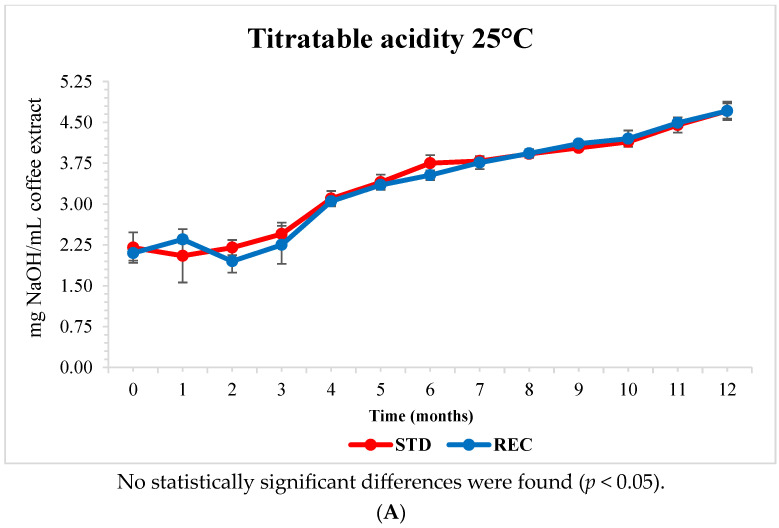
Titratable acidity (mg NaOH/mL) of coffee brew from pods packed in standard (STD) and recyclable (REC) multilayer films stored at 25 °C (**A**) and 40 °C (**B**) for 12 months.

**Table 1 molecules-29-03006-t001:** Percentage (%) of moisture in coffee pods packaged with standard (STD) and recyclable (REC) multilayer films and stored at 25 and 40 °C for 12 months.

	25 °C		40 °C	
Time (Months)	STD	REC	STD	REC
0	1.15 ± 0.09	1.21 ± 0.10	1.18 ± 0.07	1.17 ± 0.11
3	1.75 ± 0.11	1.68 ± 0.09	1.64 ± 0.09	1.56 ± 0.10
6	2.05 ± 0.13	1.99 ± 0.08	1.85 ± 0.11	1.81 ± 0.07
9	2.33 ± 0.09	2.15 ± 0.11	2.01 ± 0.06	1.98 ± 0.09
12	2.51 ± 0.10	2.37 ± 0.12	2.14 ± 0.10	2.11 ± 0.08

No statistically significant differences were found at the same time (*p* < 0.05).

**Table 2 molecules-29-03006-t002:** Relative percentage (%) of volatile organic compounds (VOCs) in coffee pods packaged with standard (STD) and recyclable (REC) multilayer films and stored at 25 °C for 12 months. ^a,b^ Different letters at the same time indicate statistically significant differences (*p* < 0.05).

Time (Months)	0	0	4	4	8	8	12	12
			STD	REC	STD	REC	STD	REC
**Σ Furans**	**43.36 ± 0.29 ^a^**	**43.38 ± 1.03 ^a^**	**41.85 ± 0.34 ^a^**	**37.14 ± 1.04 ^b^**	**35.61 ± 0.93 ^a^**	**32.65 ± 2.54 ^b^**	**37.03 ± 0.56 ^a^**	**35.44 ± 0.74 ^b^**
2-Furanmethanol	28.66 ± 0.19 ^a^	28.25 ± 0.97 ^a^	27.97 ± 0.30 ^a^	21.80 ± 1.32 ^b^	23.40 ± 0.11 ^a^	21.30 ± 0.25 ^b^	23.47 ± 0.55 ^a^	21.54 ± 0.75 ^a^
2-Furanmethanol acetate	5.18 ± 0.10 ^a^	5.32 ± 0.05 ^a^	5.13 ± 0.03 ^c^	6.43 ± 0.09 ^a^	4.69 ± 0.44 ^a^	5.16 ± 0.14 ^a^	5.2 ± 0.07 ^a^	4.12 ± 0.10 ^b^
Furfural	3.03 ± 0.03 ^b^	3.63 ± 0.01 ^a^	2.91 ± 0.26 ^a^	2.69 ± 0.03 ^a^	2.77 ± 0.14 ^a^	3.09 ± 0.06 ^a^	2.67 ± 0.08 ^b^	3.82 ± 0.02 ^a^
Dihydro-2-methyl-3-furanone	0.38 ± 0.01 ^a^	0.39 ± 0.02 ^a^	0.48 ± 0.14 ^a^	0.73 ± 0.06 ^a^	0.39 ± 0.06 ^a^	0.52 ± 0.11 ^a^	0.56 ± 0.02 ^a^	0.53 ± 0.04 ^a^
5-methyl-Furfural	3.10 ± 0.02 ^a^	3.10 ± 0.03 ^a^	3.11 ± 0.13 ^a^	3.12 ± 0.02 ^a^	2.24 ± 0.83 ^a^	0.34 ± 0.02 ^a^	3.17 ± 0.06 ^a^	2.51 ± 0.06 ^b^
2,2′-Methylenebisfuran	2.34 ± 0.01 ^a^	2.07 ± 0.03 ^b^	1.51 ± 0.06 ^b^	1.71 ± 0.21 ^b^	1.26 ± 0.01 ^a^	1.38 ± 0.14 ^a^	1.50 ± 0.05 ^b^	2.31 ± 0.26 ^a^
2-Furancarboxylic acid	0.39 ± 0.02 ^b^	0.46 ± 0.02 ^a^	0.30 ± 0.02 ^a^	0.37 ± 0.06 ^a^	0.39 ± 0.13 ^a^	0.52 ± 0.09 ^a^	0.30 ± 0.02 ^a^	0.34 ± 0.01 ^a^
Furan, 2-(methoxymethyl)-	0.28 ± 0.03 a	0.16 ± 0.01 ^b^	0.44 ± 0.10 ^a^	0.29 ± 0.01 ^a^	0.47 ± 0.11 ^a^	0.34 ± 0.18 ^a^	0.16 ± 0.02 ^a^	0.27 ± 0.07 ^a^
**Σ Pyrazines**	**30.10 ± 0.24 ^a^**	**30.34 ± 0.83 ^a^**	**31.25 ± 1.43 ^a^**	**33.19 ± 0.51 ^a^**	**33.42 ± 0.22 ^a^**	**33.25 ± 0.59 ^a^**	**31.22 ± 0.47 ^a^**	**30.12 ± 0.52 ^a^**
2-Methylpyrazine	4.95 ± 0.04 ^b^	5.54 ± 0.07 ^a^	5.91 ± 1.43 a	6.00 ± 0.10 ^a^	6.60 ± 0.08 ^a^	5.53 ± 0.11 ^c^	5.71 ± 1.71 ^a^	4.49 ± 0.49 ^a^
2,5 dimethylpyrazine	8.33 ± 0.02 ^b^	9.34 ± 0.10 ^a^	8.42 ± 0.23 ^b^	9.95 ± 0.40 ^a^	8.78 ± 0.17 ^a^	9.42 ± 0.27 ^a^	9.57 ± 0.01 ^a^	8.6 ± 0.25 ^b^
ethyl pyrazine	3.43 ± 0.40 ^a^	3.34 ± 0.24 ^a^	3.64 ± 0.13 ^a^	3.56 ± 0.14 ^a^	3.42 ± 0.40 ^a^	3.12 ± 0.08 ^a^	3.64 ± 0.13 ^a^	3.31 ± 0.23 ^a^
3,5 dimethylpyrazine	0.60 ± 0.07 ^a^	0.39 ± 0.08 ^a^	0.72 ± 0.09 ^a^	0.72 ± 0.10 ^a^	0.82 ± 0.01 ^a^	0.81 ± 0.01 ^a^	0.84 ± 0.07 ^a^	0.83 ± 0.08 ^a^
2-Ethyl-6-methylpyrazine	2.83 ± 0.09 ^a^	2.84 ± 0.01 ^a^	2.65 ± 0.34 ^b^	3.84 ± 0.04 ^a^	3.17 ± 0.54 ^a^	3.79 ± 0.34 ^a^	3.01 ± 0.85 ^a^	2.53 ± 0.11 ^a^
2,3,5-Trimethylpyrazine	4.08 ± 0.12 ^a^	2.97 ± 0.46 ^a^	4.73 ± 0.23 ^a^	3.42 ± 0.13 ^b^	4.96 ± 0.10 ^a^	5.07 ± 0.06 ^a^	3.26 ± 0.10 ^b^	4.71 ± 0.12 ^a^
3-Methoxy-2-isopropylpyrazine	1.26 ± 0.02 ^a^	1.27 ± 0.07 ^a^	1.20 ± 0.11 ^a^	0.86 ± 0.06 ^b^	1.06 ± 0.07 ^ab^	1.01 ± 0.01 ^b^	1.07 ± 0.06 ^a^	1.17 ± 0.06 ^a^
2,5-Dimethyl-3-ethylpyrazine	2.80 ± 0.02 ^b^	2.93 ± 0.01 ^a^	2.76 ± 0.05 ^a^	3.21 ± 0.21 ^a^	2.91 ± 0.26 ^a^	2.87 ± 0.18 ^a^	2.68 ± 0.07 ^a^	2.76 ± 0.08 ^a^
Isopropenylpyrazine	0.82 ± 0.05 ^a^	0.82 ± 0.01 ^a^	0.74 ± 0.08 ^a^	0.83 ± 0.11 ^a^	0.80 ± 0.03 ^a^	0.91 ± 0.09 ^a^	0.97 ± 0.41 ^a^	0.74 ± 0.08 ^a^
5-Methyl-6,7dihydro5-Hcyclopentapyrazine	0.63 ± 0.03 ^a^	0.63 ± 0.01 ^a^	0.29 ± 0.06 ^a^	0.45 ± 0.15 ^a^	0.41 ± 0.02 ^a^	0.39 ± 0.12 ^a^	0.28 ± 0.04 ^b^	0.41 ± 0.03 ^a^
2-methyl-5-(1-propenyl) Pyrazine	0.37 ± 0.04 ^ab^	0.27 ± 0.01 ^b^	0.19 ± 0.04 ^a^	0.35 ± 0.07 ^a^	0.49 ± 0.16 ^a^	0.33 ± 0.16 ^a^	0.19 ± 0.03 ^b^	0.565 ± 0.05 ^a^
**Σ Pyridines**	**4.86 ± 0.14 ^b^**	**5.62 ± 0.11 ^a^**	**6.32 ± 0.10 ^a^**	**5.32 ± 0.43 ^b^**	**6.28 ± 0.12 ^a^**	**5.52 ± 0.29 ^a^**	**6.08 ± 0.45 ^a^**	**5.39 ± 0.13 ^a^**
Pyridine	3.60 ± 0.16 ^b^	4.30 ± 0.12 ^a^	5.53 ± 0.55 ^a^	4.62 ± 0.09 ^a^	5.18 ± 0.02 ^a^	4.48 ± 0.24 ^b^	5.52 ± 0.56 ^a^	4.59 ± 0.26 ^a^
Pyridine, 1,2,3,6-tetrahydro-1-methyl-	0.95 ± 0.01 ^c^	1.00 ± 0.01 ^b^	0.58 ± 0.45 ^a^	0.39 ± 0.08 ^a^	0.72 ± 0.04 ^a^	0.62 ± 0.11 ^a^	0.36 ± 0.13 ^a^	0.48 ± 0.11 ^a^
Pyridine, 3-ethyl	0.31 ± 0.08 ^a^	0.32 ± 0.01 ^a^	0.21 ± 0.01 ^b^	0.31 ± 0.03 ^a^	0.38 ± 0.10 ^a^	0.42 ± 0.05 ^a^	0.2 ± 0.01 ^b^	0.32 ± 0.02 ^a^
**Σ Ketones**	**5.82 ± 0.49 ^a^**	**6.56 ± 0.28 ^a^**	**5.55 ± 0.34 ^b^**	**7.23 ± 0.39 ^a^**	**6.02 ± 0.65 ^a^**	**7.16 ± 0.89 ^a^**	**7.08 ± 0.49 ^a^**	**6.97 ± 0.22 ^a^**
Acetone	0.66 ± 0.01 ^b^	1.11 ± 0.01 ^a^	0.89 ± 0.04 ^a^	0.89 ± 0.08 ^a^	1.17 ± 0.01 ^a^	1.18 ± 0.06 ^a^	1.33 ± 0.01 ^b^	1.48 ± 0.06 ^a^
1-(Acetyloxy)-2-propanone	3.42 ± 0.23 ^a^	3.92 ± 0.01 ^a^	3.23 ± 0.06 ^b^	5.06 ± 0.20 ^a^	3.96 ± 0.57 ^a^	4.57 ± 0.32 ^a^	4.16 ± 0.04 ^a^	4.03 ± 0.04 ^a^
2-Hydroxy-3-methyl-2- cyclopenten-1-one	0.49 ± 0.01 ^b^	0.47 ± 0.03 ^a^	0.52 ± 0.11 ^a^	0.34 ± 0.02 ^a^	0.40 ± 0.06 ^a^	0.49 ± 0.19 ^a^	0.49 ± 0.08 ^a^	0.46 ± 0.06 ^a^
3-Cyclobutene-1,2-dione, 3,4-dimethyl-	0.66 ± 0.03 ^a^	0.55 ± 0.03 ^b^	0.41 ± 0.06 ^a^	0.42 ± 0.05 ^a^	0.36 ± 0.01 ^b^	0.43 ± 0.10 ^ab^	0.35 ± 0.02 ^b^	0.65 ± 0.08 ^a^
3-Ethyl-2-hydroxy-2- Cyclopenten-1-one	0.59 ± 0.02 ^ab^	0.51 ± 0.01 ^a^	0.50 ± 0.08 ^a^	0.52 ± 0.09 ^a^	0.40 ± 0.03 ^a^	0.49 ± 0.10 ^a^	0.75 ± 0.42 ^a^	0.35 ± 0.07 ^a^
**Σ Phenols**	**7.97 ± 0.45 ^a^**	**5.69 ± 0.19 ^b^**	**6.78 ± 0.38 ^a^**	**6.34 ± 0.55 ^a^**	**7.95 ± 0.13 ^b^**	**9.06 ± 0.25 ^a^**	**5.64 ± 1.13 ^a^**	**6.29 ± 0.17 ^a^**
Phenol	0.25 ± 0.01 ^a^	0.29 ± 0.03 ^a^	0.29 ± 0.04 ^b^	0.77 ± 0.18 ^a^	0.50 ± 0.22 ^a^	1.37 ± 0.40 ^a^	0.30 ± 0.02 ^ab^	0.43 ± 0.04 ^a^
2-methoxyphenol	3.40 ± 0.36 ^a^	1.65 ± 0.01 ^b^	3.63 ± 0.19 ^a^	1.84 ± 0.13 ^b^	3.33 ± 0.01 ^a^	3.82 ± 0.11 ^a^	2.32 ± 0.77 ^a^	3.75 ± 0.06 ^a^
4-ethyl-2-methoxyphenol	1.67 ± 0.06 ^ab^	1.46 ± 0.11 ^b^	1.18 ± 0.10 ^a^	1.61 ± 0.17 ^a^	1.36 ± 0.25 ^a^	1.68 ± 0.29 ^a^	1.24 ± 0.10 ^a^	1.04 ± 0.04 ^a^
4-Vinylphenol	2.65 ± 0.01 ^b^	2.29 ± 0.06 ^c^	1.68 ± 0.13 ^a^	2.12 ± 0.07 ^a^	2.76 ± 0.16 ^a^	2.19 ± 0.06 ^a^	1.78 ± 0.28 ^a^	1.07 ± 0.07 ^b^
**Σ Pyrroles**	**2.54 ± 0.11 ^a^**	**2.32 ± 0.09 ^b^**	**2.28 ± 0.23 ^a^**	**3.00 ± 0.09 ^a^**	**2.60 ± 0.09 ^a^**	**2.65 ± 0.10 ^a^**	**2.22 ± 0.14 ^a^**	**2.25 ± 0.15 ^a^**
2-Acetylpyrrole	0.55 ± 0.02 ^a^	0.59 ± 0.01 ^a^	0.68 ± 0.02 ^c^	1.07 ± 0.02 ^a^	0.70 ± 0.05 ^a^	0.68 ± 0.09 ^a^	0.72 ± 0.08 ^a^	0.79 ± 0.05 ^a^
2-Acetyl-1-methylpyrrole	0.72 ± 0.02 ^a^	0.66 ± 0.01 ^a^	0.61 ± 0.04 ^a^	0.72 ± 0.07 ^a^	0.61 ± 0.06 ^a^	0.65 ± 0.01 ^a^	0.61 ± 0.05 ^a^	0.75 ± 0.06 ^a^
1H-Pyrrole, 1-(2-furanylmethyl)-	1.27 ± 0.03 ^a^	1.07 ± 0.01 ^b^	0.99 ± 0.16 ^a^	1.21 ± 0.04 ^a^	1.29 ± 0.02 ^a^	1.32 ± 0.01 ^a^	0.89 ± 0.06 ^a^	0.705 ± 0.26 ^a^
**Σ Aldehydes**	**0.32 ± 0.04 ^b^**	**0.40 ± 0.01 ^a^**	**0.32 ± 0.04 ^a^**	**0.46 ± 0.15 ^a^**	**0.44 ± 0.02 ^a^**	**0.41 ± 0.16 ^a^**	**0.32 ± 0.04 ^b^**	**0.66 ± 0.06 ^a^**
3-Methyl-p-anisaldehyde	0.32 ± 0.04 ^b^	0.40 ± 0.01 ^a^	0.32 ± 0.04 ^a^	0.46 ± 0.15 ^a^	0.44 ± 0.02 ^a^	0.41 ± 0.16 ^a^	0.32 ± 0.04 ^b^	0.66 ± 0.06 ^a^
**Σ Organic Acid**	**5.03 ± 0.11 ^b^**	**5.69 ± 0.05 ^a^**	**5.65 ± 0.28 ^b^**	**7.32 ± 0.11 ^a^**	**7.68 ± 0.11 ^b^**	**9.30 ± 0.16 ^a^**	**10.41 ± 0.25 ^b^**	**12.89 ± 0.11 ^a^**
Acetic acid	4.51 ± 0.14 ^b^	5.27 ± 0.04 ^a^	5.32 ± 0.25 ^b^	6.90 ± 0.05 ^a^	7.25 ± 0.15 ^b^	8.97 ± 0.25 ^a^	10.12 ± 0.22 ^b^	12.57 ± 0.13 ^a^
3-Methylbutanoic acid	0.52 ± 0.05 ^a^	0.42 ± 0.01 ^b^	0.33 ± 0.09 ^a^	0.42 ± 0.16 ^a^	0.43 ± 0.04 ^a^	0.33 ± 0.09 ^a^	0.29 ± 0.03 ^a^	0.315 ± 0.02 ^a^

**Table 3 molecules-29-03006-t003:** Relative percentage (%) of volatile organic compounds (VOCs) in coffee pods packaged with standard (STD) and recyclable (REC) multilayer films and stored at 40 °C for 12 months. ^a,b^ Different letters at the same time indicate statistically significant differences (*p* < 0.05).

Time (Months)	0	0	4	4	8	8	12	12
			STD	REC	STD	REC	STD	REC
**Σ Furans**	**43.22 ± 1.88 ^a^**	**43.38 ± 1.03 ^a^**	**41.88 ± 0.79 ^a^**	**38.71 ± 2.50 ^a^**	**35.35 ± 0.30 ^a^**	**35.58 ± 0.16 ^a^**	**36.15 ± 0.70 ^a^**	**33.17 ± 0.47 ^b^**
2-Furanmethanol	28.50 ± 1.22 ^a^	28.12 ± 0.97 ^a^	25.83 ± 1.07 ^a^	23.30 ± 2.29 ^a^	23.00 ± 1.19 ^a^	22.33 ± 0.13 ^ab^	22.30 ± 0.76 ^a^	20.29 ± 0.21 ^a^
2-Furanmethanol acetate	5.18 ± 0.38 ^a^	5.32 ± 0.05 ^a^	4.71 ± 0.35 ^a^	5.11 ± 0.22 ^a^	4.69 ± 0.44 ^a^	4.51 ± 0.19 ^a^	5.20 ± 0.07 ^a^	4.41 ± 0.15 ^b^
Furfural	3.04 ± 0.14 ^c^	3.63 ± 0.01 ^a^	2.02 ± 0.09 ^a^	4.08 ± 0.13 ^a^	2.77 ± 0.46 ^a^	3.03 ± 0.19 ^a^	2.67 ± 0.08 ^a^	2.50 ± 0.16 ^a^
Dihydro-2-methyl-3-furanone	0.38 ± 0.01 ^a^	0.39 ± 0.02 ^a^	0.60 ± 0.02 ^b^	0.66 ± 0.15 ^a^	0.39 ± 0.02 ^a^	0.40 ± 0.08 ^a^	0.56 ± 0.02 ^a^	0.50 ± 0.07 ^a^
5-methyl-Furfural	3.11 ± 0.11 ^a^	3.10 ± 0.03 ^a^	5.11 ± 0.31 ^a^	3.19 ± 0.31 ^b^	2.24 ± 1.01 ^a^	2.51 ± 0.01 ^a^	3.17 ± 0.06 ^a^	2.71 ± 0.13 ^b^
2,2′-Methylenebisfuran	2.34 ± 0.07 ^a^	2.07 ± 0.03 ^b^	2.61 ± 0.08 ^a^	1.59 ± 0.06 ^b^	1.26 ± 0.02 ^a^	1.78 ± 0.74 ^a^	1.50 ± 0.21 ^a^	2.02 ± 0.67 ^a^
2-Furancarboxylic acid	0.40 ± 0.01 ^ab^	0.46 ± 0.02 ^a^	0.66 ± 0.03 ^a^	0.27 ± 0.04 ^a^	0.39 ± 0.13 ^a^	0.32 ± 0.02 ^a^	0.30 ± 0.02 ^a^	0.34 ± 0.06 ^a^
Furan, 2-(methoxymethyl)-	0.28 ± 0.02 ^a^	0.16 ± 0.01 ^b^	0.35 ± 0.00 ^a^	0.35 ± 0.02 ^a^	0.39 ± 0.06 ^a^	0.55 ± 0.15 ^a^	0.37 ± 0.04 ^a^	0.30 ± 0.01 ^b^
**Σ Pyrazines**	**30.15 ± 0.55 ^a^**	**30.34 ± 0.83 ^a^**	**28.21 ± 0.25 ^b^**	**31.35 ± 0.47 ^a^**	**32.86 ± 0.17 ^a^**	**32.42 ± 0.44 ^a^**	**31.87 ± 0.78 ^a^**	**29.02 ± 0.22 ^b^**
2-Methylpyrazine	4.95 ± 0.12 ^b^	5.54 ± 0.07 ^a^	5.37 ± 0.24 ^a^	5.92 ± 0.65 ^a^	6.60 ± 0.28 ^a^	5.68 ± 0.75 ^a^	5.71 ± 0.71 ^a^	4.31 ± 0.26 ^a^
2,5 dimethylpyrazine	8.34 ± 0.34 ^b^	9.34 ± 0.10 ^a^	8.11 ± 0.51 ^b^	9.28 ± 0.41 ^a^	8.40 ± 0.54 ^a^	9.55 ± 0.08 ^a^	9.32 ± 0.35 ^a^	9.27 ± 0.24 ^a^
ethyl pyrazine	3.43 ± 0.09 ^a^	3.34 ± 0.24 ^a^	3.47 ± 0.11 ^a^	4.12 ± 0.54 ^a^	3.42 ± 0.33 ^a^	3.14 ± 0.03 ^a^	3.64 ± 0.13 ^a^	3.71 ± 0.06 ^a^
3,5 dimethylpyrazine	0.61 ± 0.03 ^a^	0.39 ± 0.08 ^a^	0.27 ± 0.01 ^a^	0.39 ± 0.06 ^a^	0.82 ± 0.09 ^a^	0.45 ± 0.01 ^b^	0.84 ± 0.07 ^a^	0.79 ± 0.01 ^a^
2-Ethyl-6-methylpyrazine	2.83 ± 0.09 ^a^	2.84 ± 0.01 ^a^	2.86 ± 0.08 ^b^	3.03 ± 0.01 ^b^	3.17 ± 0.54 ^a^	2.90 ± 0.29 ^a^	3.36 ± 0.35 ^a^	2.32 ± 0.15 ^b^
2,3,5-Trimethylpyrazine	4.08 ± 0.21 ^a^	2.97 ± 0.46 ^a^	3.49 ± 0.13 ^b^	3.85 ± 0.48 ^a^	4.96 ± 0.20 ^a^	4.69 ± 0.54 ^a^	3.26 ± 0.10 ^a^	3.10 ± 0.66 ^a^
3-Methoxy-2-isopropylpyrazine	1.26 ± 0.05 ^a^	1.27 ± 0.07 ^a^	0.98 ± 0.04 ^a^	1.12 ± 0.17 ^a^	1.06 ± 0.11 ^a^	1.20 ± 0.23 ^a^	1.07 ± 0.06 ^a^	1.20 ± 0.11 ^a^
2,5-Dimethyl-3-ethylpyrazine	2.81 ± 0.12 ^b^	2.93 ± 0.01 ^a^	1.94 ± 0.06 ^a^	2.09 ± 0.03 ^a^	2.91 ± 0.20 ^a^	3.05 ± 0.07 ^a^	2.68 ± 0.77 ^a^	2.63 ± 0.27 ^a^
Isopropenylpyrazine	0.82 ± 0.03 ^a^	0.82 ± 0.01 ^a^	0.45 ± 0.02 ^a^	0.67 ± 0.09 ^a^	0.80 ± 0.29 ^a^	0.85 ± 0.05 ^a^	0.97 ± 0.06 ^a^	0.80 ± 0.01 ^a^
5-Methyl-6,7dihydro5- Hcyclopentapyrazine	0.64 ± 0.02 ^a^	0.63 ± 0.01 ^a^	0.87 ± 0.03 ^a^	0.39 ± 0.05 ^b^	0.29 ± 0.14 ^a^	0.47 ± 0.14 ^a^	0.73 ± 0.06 ^a^	0.41 ± 0.13 ^b^
2-methyl-5-(1-propenyl) Pyrazine	0.38 ± 0.01 ^ab^	0.27 ± 0.01 ^b^	0.41 ± 0.01 ^a^	0.49 ± 0.11 ^a^	0.43 ± 0.07 ^a^	0.44 ± 0.07 ^a^	0.29 ± 0.06 ^a^	0.48 ± 0.07 ^a^
**Σ Pyridines**	**4.88 ± 0.15 ^b^**	**5.62 ± 0.11 ^a^**	**5.99 ± 0.21 ^a^**	**3.68 ± 0.02 ^b^**	**6.28 ± 0.18 ^a^**	**6.45 ± 0.69 ^a^**	**6.08 ± 0.45 ^a^**	**5.33 ± 0.54 ^a^**
Pyridine	3.61 ± 0.05 ^b^	4.30 ± 0.12 ^a^	4.84 ± 0.12 ^a^	2.59 ± 0.04 ^b^	5.18 ± 0.02 ^a^	5.25 ± 0.57 ^a^	5.52 ± 0.56 ^a^	4.51 ± 0.54 ^a^
Pyridine, 1,2,3,6-tetrahydro-1-methyl-	0.95 ± 0.04 ^b^	1.00 ± 0.01 ^b^	0.45 ± 0.01 ^a^	0.59 ± 0.15 ^a^	0.72 ± 0.06 ^a^	0.82 ± 0.09 ^a^	0.36 ± 0.13 ^a^	0.50 ± 0.08 ^a^
Pyridine, 3-ethyl	0.32 ± 0.01 ^a^	0.32 ± 0.01 ^a^	0.70 ± 0.02 ^a^	0.50 ± 0.06 ^ab^	0.38 ± 0.11 ^a^	0.38 ± 0.03 ^a^	0.20 ± 0.01 ^a^	0.32 ± 0.08 ^a^
**Σ Ketones**	**5.85 ± 0.32 ^a^**	**6.56 ± 0.08 ^a^**	**6.86 ± 0.44 ^a^**	**7.82 ± 0.18 ^a^**	**6.30 ± 0.24 ^a^**	**6.41 ± 0.92 ^a^**	**6.61 ± 0.11 ^b^**	**6.94 ± 0.18 ^a^**
Acetone	0.66 ± 0.03 ^b^	1.11 ± 0.01 ^a^	0.72 ± 0.03 ^a^	0.73 ± 0.03 ^a^	1.17 ± 0.01 ^a^	1.19 ± 0.01 ^a^	1.33 ± 0.01 ^b^	1.55 ± 0.07 ^a^
1-(Acetyloxy)-2-propanone	3.42 ± 0.12 ^a^	3.92 ± 0.01 ^a^	4.01 ± 0.25 ^b^	5.09 ± 0.05 ^a^	3.96 ± 0.44 ^a^	3.90 ± 0.15 ^a^	4.16 ± 0.04 ^a^	3.86 ± 0.40 ^ab^
2-Hydroxy-3-methyl-2- cyclopenten-1-one	0.50 ± 0.01 ^b^	0.47 ± 0.03 a	0.61 ± 0.03 ^a^	0.68 ± 0.03 ^a^	0.40 ± 0.01 ^a^	0.46 ± 0.17 ^a^	0.49 ± 0.08 ^a^	0.51 ± 0.13 ^a^
3-Cyclobutene-1,2-dione, 3,4-dimethyl-	0.67 ± 0.02 ^a^	0.55 ± 0.03 ^b^	0.90 ± 0.04 ^a^	0.66 ± 0.04 ^b^	0.36 ± 0.10 ^a^	0.45 ± 0.18 ^a^	0.35 ± 0.02 ^a^	0.53 ± 0.10 ^a^
3-Ethyl-2-hydroxy-2- Cyclopenten-1-one	0.60 ± 0.02 ^a^	0.51 ± 0.01 ^b^	0.63 ± 0.01 ^a^	0.66 ± 0.16 ^a^	0.41 ± 0.01 ^a^	0.41 ± 0.01 ^a^	0.28 ± 0.04 ^a^	0.49 ± 0.13 ^a^
**Σ Phenols**	**7.99 ± 0.43 ^a^**	**5.69 ± 0.09 ^b^**	**7.95 ± 0.51 ^a^**	**6.75 ± 1.20 ^a^**	**8.30 ± 0.66 ^a^**	**6.14 ± 0.27 ^b^**	**6.24 ± 0.47 ^a^**	**7.31 ± 0.38 ^a^**
Phenol	0.26 ± 0.00 ^a^	0.29 ± 0.03 ^a^	0.48 ± 0.03 ^a^	0.96 ± 0.07 ^a^	0.50 ± 0.22 ^a^	0.36 ± 0.04 ^a^	0.30 ± 0.02 ^b^	0.42 ± 0.04 ^a^
2-methoxyphenol	3.41 ± 0.15 ^a^	1.65 ± 0.01 ^b^	2.86 ± 0.07 ^a^	2.30 ± 0.18 ^a^	3.33 ± 1.00 ^a^	3.24 ± 0.13 ^a^	2.32 ± 0.04 ^a^	3.78 ± 0.02 ^a^
4-ethyl-2-methoxyphenol	1.68 ± 0.08 ^ab^	1.46 ± 0.11 ^b^	1.51 ± 0.11 ^a^	1.85 ± 0.19 ^a^	2.21 ± 0.40 ^a^	1.18 ± 0.05 ^b^	1.31 ± 0.14 ^a^	1.58 ± 0.11 ^a^
4-Vinylphenol	2.65 ± 0.14 ^a^	2.29 ± 0.06 ^b^	3.11 ± 0.28 ^a^	1.64 ± 0.38 ^b^	2.26 ± 0.27 ^a^	1.36 ± 0.05 ^b^	2.31 ± 0.12 ^a^	1.53 ± 0.33 ^a^
**Σ Pyrroles**	**2.56 ± 0.08 ^a^**	**2.32 ± 0.01 ^b^**	**2.85 ± 0.15 ^a^**	**3.28 ± 0.06 ^a^**	**2.79 ± 0.06 ^a^**	**2.76 ± 0.14 ^a^**	**2.49 ± 0.19 ^a^**	**2.23 ± 0.13 ^a^**
2-Acetylpyrrole	0.55 ± 0.02 ^a^	0.59 ± 0.01 ^a^	0.33 ± 0.02 ^b^	0.78 ± 0.04 ^a^	0.70 ± 0.08 ^a^	0.66 ± 0.08 ^a^	0.72 ± 0.08 ^a^	0.76 ± 0.09 ^a^
2-Acetyl-1-methylpyrrole	0.73 ± 0.03 ^a^	0.66 ± 0.01 ^a^	1.26 ± 0.09 ^a^	0.86 ± 0.01 ^b^	0.61 ± 0.11 ^a^	0.72 ± 0.10 ^a^	0.61 ± 0.05 ^a^	0.72 ± 0.11 ^a^
1H-Pyrrole, 1-(2-furanylmethyl) -	1.28 ± 0.06 ^a^	1.07 ± 0.01 ^b^	1.26 ± 0.08 ^a^	1.64 ± 0.61 ^a^	1.48 ± 0.08 ^a^	1.38 ± 0.08 ^a^	1.16 ± 0.01 ^a^	0.75 ± 0.23 ^a^
**Σ Aldehydes**	**0.33 ± 0.02 ^b^**	**0.40 ± 0.01 ^a^**	**0.48 ± 0.03 ^a^**	**0.62 ± 0.03 ^a^**	**0.49 ± 0.04 ^a^**	**0.38 ± 0.06 ^a^**	**0.19 ± 0.04 ^a^**	**0.53 ± 0.23 ^a^**
3-Methyl-p-anisaldehyde	0.33 ± 0.01 ^b^	0.40 ± 0.01 ^a^	0.48 ± 0.02 ^a^	0.62 ± 0.04 ^a^	0.49 ± 0.04 ^a^	0.38 ± 0.04 ^a^	0.19 ± 0.03 ^a^	0.53 ± 0.01 ^a^
**Σ Organic Acid**	**5.03 ± 0.19 ^b^**	**5.69 ± 0.05 ^a^**	**5.79 ± 0.26 ^b^**	**7.79 ± 0.03 ^a^**	**7.63 ± 0.22 ^b^**	**9.86 ± 0.32 ^a^**	**10.37 ± 0.28 ^b^**	**15.47 ± 0.39 ^a^**
Acetic acid	4.51 ± 0.18 ^b^	5.27 ± 0.04 ^a^	5.32 ± 0.38 ^b^	7.32 ± 0.03 ^a^	7.25 ± 0.15 ^b^	9.34 ± 0.28 ^a^	10.12 ± 0.22 ^b^	15.10 ± 0.42 ^a^
3-Methylbutanoic acid	0.52 ± 0.04 ^a^	0.42 ± 0.01 ^b^	0.47 ± 0.01 ^a^	0.47 ± 0.11 ^a^	0.38 ± 0.03 ^a^	0.52 ± 0.04 ^a^	0.25 ± 0.11 ^a^	0.37 ± 0.04 ^a^

**Table 4 molecules-29-03006-t004:** pH values of coffee brew from pods packed in standard multilayer film (STD) and recyclable film (REC) and stored at 25 °C and 40 °C for 12 months.

	25 °C		40 °C	
Time (Months)	STD	REC	STD	REC
0	5.66 ± 0.02 ^a^	5.62 ± 0.02 ^b^	5.66 ± 0.02 ^a^	5.62 ± 0.02 ^b^
1	5.67 ± 0.03 ^a^	5.68 ± 0.02 ^a^	5.64 ± 0.02 ^a^	5.63 ± 0.02 ^a^
2	5.65 ± 0.04 ^a^	5.50 ± 0.02 ^b^	5.65 ± 0.02 ^a^	5.54 ± 0.03 ^b^
3	5.63 ± 0.02 ^a^	5.61 ± 0.02 ^a^	5.62 ± 0.02 ^a^	5.52 ± 0.05 ^b^
4	5.55 ± 0.02 ^a^	5.59 ± 0.04 ^a^	5.54 ± 0.02 ^a^	5.48 ± 0.02 ^b^
5	5.54 ± 0.02 ^a^	5.55 ± 0.03 ^a^	5.46 ± 0.02 ^a^	5.40 ± 0.02 ^b^
6	5.53 ± 0.02 ^a^	5.52 ± 0.03 ^a^	5.40 ± 0.02 ^a^	5.37 ± 0.02 ^a^
7	5.48 ± 0.02 ^a^	5.47 ± 0.02 ^a^	5.36 ± 0.02 ^a^	5.35 ± 0.02 ^a^
8	5.48 ± 0.02 ^a^	5.46 ± 0.02 ^a^	5.31 ± 0.02 ^a^	5.32 ± 0.02 ^a^
9	5.46 ± 0.02 ^a^	5.44 ± 0.02 ^a^	5.29 ± 0.05 ^a^	5.31 ± 0.02 ^a^
10	5.44 ± 0.02 ^a^	5.44 ± 0.02 ^a^	5.29 ± 0.04 ^a^	5.29 ± 0.03 ^a^
11	5.39 ± 0.02 ^a^	5.41 ± 0.02 ^a^	5.30 ± 0.02 ^a^	5.30 ± 0.02 ^a^
12	5.37 ± 0.02 ^a^	5.32 ± 0.02 ^a^	5.29 ± 0.02 ^a^	5.28 ± 0.02 ^a^

^a,b^ Different letters at the same time and temperature indicate statistically significant differences (*p* < 0.05).

**Table 5 molecules-29-03006-t005:** Smell and taste acceptability of coffee brew from pods packaged with standard (STD) and recyclable (REC) multilayer films and stored at 25 °C for 12 months.

	STD	REC	STD	REC	STD	REC	STD	REC
Time (Months)	0	0	4	4	8	8	12	12
Smell acceptability	8.5 ± 0.5 ^a^	8.5 ± 0.5 ^a^	8.5 ± 0.5 ^a^	8.5 ± 0.5 ^a^	8.0 ± 0.5 ^a^	8.0 ± 0.5 ^a^	7.5 ± 0.5 ^b^	7.5 ± 0.5 ^b^
Taste acceptability	8.5 ± 0.5 ^a^	8.5 ± 0.5 ^a^	8.5 ± 0.5 ^a^	8.5 ± 0.5 ^a^	8.0 ± 0.5 ^a^	8.0 ± 0.5 ^a^	7.5 ± 0.5 ^b^	7.5 ± 0.5 ^b^

^a,b^ Different letters at the same time indicate statistically significant differences (*p* < 0.05).

## Data Availability

Data are contained within the article.

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
