# Peer review of "A Recyclable Polypropylene Multilayer Film Maintaining the Quality and the Aroma of Coffee Pods during Their Shelf Life"

_molecules, 2024, doi:10.3390/molecules29133006_

Round 1

Reviewer 1 Report

Comments and Suggestions for Authors

Thank you for the opportunity to read this interesting publication: “Recyclable polypropylene multilayer film maintains the quality and the aroma of coffee pods during the shelf life.”

The authors did a lot of work and were very thorough in their approach to the study. They developed multilayer polypropylene films for packaging coffee capsules and assessed the effect of packaging on coffee quality. The article is interesting, but there are some points that could improve the publication.

1. The publication contains a large percentage of borrowings (34%). Although the bulk of borrowing occurs in the materials and methods section, authors should not use copy-paste functions when preparing a publication.

2. Although the journal does not strictly regulate the structure of the publication, and the section “materials and methods” may be located at the end, the abbreviations used by the authors should be disclosed at the first mention in the text, and not at the end of the publication.

3. Photos of packaged capsules should be provided.

4. Authors often use the word eco-friendly to describe polypropylene packaging, but, unfortunately, polypropylene is hardly an environmentally friendly material. Recyclable yes, but environmentally friendly no.

5. The use of PP for sealing coffee capsules cannot be called an innovation. For example, the company cartedozio offers such films. https://www.cartedozio.com/en/carta-filtro-per-macchine-riempitrici/capsule/. https://www.almasa.ch/en/coffee-capsules-from-sustainable-film/.

Authors should analyze similar packaging in the introduction and explain how their development differs from analogues.

Reviewer 2 Report

Comments and Suggestions for Authors

Dear Authors,

The Authors examined the influence of the packaging material on selected parameters of coffee stored in sachets. They examined all the most critical parameters necessary to determine the quality of coffee. The results showed that the REC material can successfully replace the previously used metalized material. From this point of view, the authors achieved the article's purpose.

It is debatable whether such test results could not have been predicted, considering the REC and STD properties as the packaging materials. REC's barrier to oxygen is higher than STD, so the lower oxygen content in the packaging is understandable (Fig. 1). Why should other properties of a packaged product change differently?

Are the Authors aware of any research on the barrier properties of packaging materials concerning volatile organic compounds?

Will new EU regulations introducing restrictions on single-use plastic packaging for fruit and vegetables, food and drinks, spices, and sauces in the hospitality sector make your work less important?

Detailed comments

line 123

The article's text states that oxygen concentration in the headspace achieves a maximum of 0.93 and 1% after 12 months of storage. In Figure 1, the oxygen concentration in both cases equals 1%. Which data is correct?

I suggest introducing a graph of the percentage of oxygen in the headspace at temperaturÄ™ 40. All other figures and tables are made for 25 and 40C.

line 167

REC does not have barrier properties against oxidation processes. However, oxidation processes occur to a lesser extent due to its better barrier properties towards oxygen.

line 363

The methodology mentioned that the pH measurement was performed using the Meditor pH BASIC 20 Crison Instruments device. What buffers were used? Usually, the pH accuracy of the buffer is +-0.02. So, does it make sense to provide pH results with an accuracy of 0.01?

 The sachets are closed by heat sealing. Has the tightness of the closure been tested?

What was the volume of the sachet? I didn't find this information in the article. Was the volume sufficient to measure oxygen content?
